

*Manuscript for*

# Divergent Drivers of Aerosol Acidity: Evidence for Shifting Regulatory Regimes in a Coastal Region

**Authors**: Jinghao Zhai[1,2], Yujie Zhang[1], Baohua Cai[1], Yaling Zeng[1,2], Jingyi Zhang[1], Jianhuai
Ye[1,2], Chen Wang[1,2], Tzung-May Fu[1,2], Lei Zhu[1,2], Huizhong Shen[1,2], Xin Yang[1,2]*
*[1]Shenzhen Key Laboratory of Precision Measurement and Early Warning Technology for Urban*
*Environmental Health Risks, School of Environmental Science and Engineering, Southern*
*University of Science and Technology, Shenzhen 518055, China*
*[2]Guangdong Provincial Observation and Research Station for Coastal Atmosphere and Climate*
*of the Greater Bay Area, Shenzhen 518055, China*
*To whom correspondence should be addressed.
Correspondence to: Xin Yang
Email: yangx@sustech.edu.cn





**ABSTRACT:** Aerosol acidity plays a crucial role in multiphase atmospheric chemistry,
influencing aerosol composition, gas-particle partitioning, and the oxidative capacity of
atmosphere. However, the mechanisms governing aerosol acidity in coastal area under extreme
weather remains challenging due to its complexity of atmospheric transport. Here, we investigate
aerosol pH in Shenzhen, a coastal megacity in China, by integrating field observations with
multiphase buffer theory and ISORROPIA simulations. Our observations captured both a typhoon
episode and typical non-typhoon periods with two contrasting regimes: during non-typhoon
periods, aerosols were consistently buffered by the $NH_4^+/NH_3$ pair, with relative humidity serving
as the primary driver of pH variability, enabling reliable predictions using multiphase buffer theory.
In contrast, during a typhoon episode, sea salt derived nonvolatile cations emerged as the dominant
drivers, violating the charge balance for $NH_4^+/NH_3$ buffering and leading to poor performance of
buffer theory. Under these conditions, ISORROPIA simulations with constant aerosol water
content reproduced the observed pH more reliably, highlighting a compositional rather than
meteorological control. Our results provide the direct field-based evidence for regime shifts in
aerosol acidity regulation in coastal area, and underscore the need for chemical transport models
to account for composition-meteorology interactions to improve acidity predictions under extreme
weather events.
**Abstract Graphic**

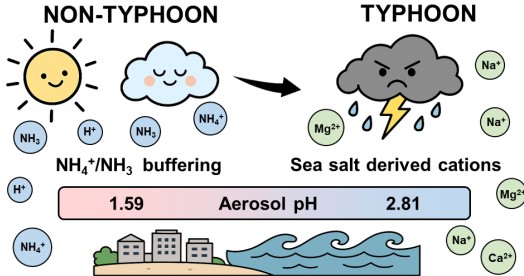




## 1 INTRODUCTION

Aerosol acidity is a key regulator in multiphase atmospheric chemistry. It governs the gas-particle partitioning of semi-volatile species (e.g., $NH_3$, $HNO_3$, $HCl$, and organic acids/bases) and dictates critical aqueous-phase processes, including $SO_2$ oxidation, secondary transformations of organic compounds, and the activation of trace metals (Tilgner et al., 2021;Pye et al., 2020;Cai et al., 2024;Surratt et al., 2007). Through these pathways, aerosol acidity exerts strong control over atmospheric oxidative capacity and pollutant lifetimes (Pye et al., 2020). On a large scale, aerosol acidity plays a pivotal role in determining particle composition, optical properties, and hygroscopicity, thereby influencing their radiative impacts and the ability to act as cloud condensation nuclei (Turnock et al., 2019;Karydis et al., 2021;Xu et al., 2020;Zhang et al., 2023). Moreover, aerosol acidity enhances particle toxicity by directly triggering respiratory inflammation, and affects the solubility of heavy metals, thereby regulating their bioavailability in terrestrial and marine ecosystems (Fang et al., 2017;Song et al., 2024;Amdur et al., 1978;Longo et al., 2016). Variations in aerosol acidity are not only fundamental to atmospheric chemistry processes but also directly influence regional air quality management (e.g., coordinated control of $PM_{2.5}$ and ozone), the global nitrogen and sulfur cycles, and climate feedback mechanisms. Thus, advancing the understanding of aerosol acidity has critical implications for public health and environmental policy.

Precise quantification of aerosol acidity remains an important yet challenging issue in atmospheric chemistry. Aerosol acidity is typically determined by aerosol pH. Traditional methods based on filter extraction and subsequent $H^+$ quantification are susceptible to substantial artifacts arising from sampling and dilution (Pathak et al., 2004;Hennigan et al., 2015). Recent techniques, such as Raman-based microdroplet pH detection (Cui et al., 2021), aerosol optical tweezers (Boyer et al., 2020), fluorescence probes (Li and Kuwata, 2023), and quantitative colorimetric imaging (Craig et al., 2018), have provided novel insights into particle-scale acidity, though their applicability to real-world aerosols remains limited, particularly for real-time dynamic monitoring of aerosol acidity. Indirect proxies including ion balance and gas-to-particle molar ratios, while



widely applied, suffer from systematic biases owing to neglected organic acid dissociation and
semi-volatile partitioning (Metzger et al., 2006;Hennigan et al., 2015). Thermodynamic
equilibrium models have emerged as the dominant framework for estimating aerosol pH values
(Saxena et al., 1986;Jacobson et al., 1996;Pilinis and Seinfeld, 1987;Wexler and Seinfeld, 1991).
Among them, the Extended Aerosol Inorganic Model (E-AIM) and ISORROPIA are widely
employed (Wexler and Clegg, 2002;Nenes et al., 1998), with E-AIM regarded as the benchmark
owing to its explicit treatment of ion activity coefficients (Clegg et al., 1992), while ISORROPIA
is favored for its computational efficiency (Nenes et al., 1999). Comparative studies generally
report consistent pH estimates from these two models, albeit with context-dependent deviations
(Song et al., 2018;Hennigan et al., 2015;Battaglia et al., 2019). Substantial uncertainties persist in
characterizing the spatiotemporal variability of aerosol acidity and its dynamic response to
chemical composition and meteorological drivers. Research remains limited in coastal megacities
and under extreme weather events such as typhoons, where intense atmospheric transport may
substantially challenge the applicability of existing models and theories.
Aerosol acidity exhibits strong spatiotemporal variability, mainly arising from the combined
influences of particle chemical composition and meteorological conditions (Zhou et al.,
2022;Zhang et al., 2021;Ding et al., 2019;Wang et al., 2022;Tao and Murphy, 2019, 2021). In
particular, water-soluble inorganic components exert significant control, with sulfate substantially
enhancing aerosol acidity due to its low volatility, whereas nitrate, with its strong hygroscopicity,
increases aerosol water content (AWC) under elevated relative humidity, thereby lowering acidity
(Ding et al., 2019). Therefore, a lower nitrate-to-sulfate ratio generally leads to more acidic
particles (Xie et al., 2020). Although the direct contribution of organics to aerosol pH is relatively
minor (Guo et al., 2015), interactions between inorganic and organic components can alter acidity
with pH increases of up to 0.7 units (Pye et al., 2018). Meteorological conditions exert a strong
influence on aerosol acidity by altering both particle water content and gas-particle partitioning.
Increased relative humidity facilitates hygroscopic growth and promotes aqueous-phase reactions,
which typically dilute proton concentrations and thus mitigate acidity (Bian et al., 2014). In



contrast, higher temperatures shift the equilibrium of semi-volatile species such as ammonia
toward the gas phase and enhance water vapor pressure, processes that together elevate proton
loading and intensify aerosol acidity (Guo et al., 2018;Battaglia et al., 2017). The recently
proposed multiphase buffer theory provides a new framework for understanding the long-term
evolution of aerosol acidity by emphasizing the stabilizing role of conjugate acid-base pairs against
external perturbations, thereby exerting critical control over sulfate formation pathways and other
multiphase atmospheric processes (Zheng et al., 2020;Zheng et al., 2022;Chen et al., 2022;Gao et
al., 2025), highlighting the complexity of aerosol acidity driving mechanisms. However, the
relative importance of different buffering systems under complex meteorological conditions
remains poorly constrained, and the potential dominance of nonvolatile cations (NVCs, e.g., $Na^+$,
$Ca^{2+}$, $Mg^{2+}$, $K^+$) in coastal environments and during extreme weather events has not been
systematically assessed with field evidence.
In this study, we investigated the buffering capacity and controlling factors of aerosol pH in
Shenzhen, China, a subtropical coastal megacity frequently influenced by typhoons, by integrating
field observations with multiphase buffer theory and ISORROPIA II simulations. Our field
observations captured both a typhoon episode and typical non-typhoon periods, providing a natural
contrast between distinct atmospheric regimes. By comparing the buffering capacity and key
drivers of aerosol pH during typhoon and non-typhoon periods, we aim to shed light on the
complex interactions between composition and meteorology that regulate aerosol acidity. Our
results provide the field-based evidence the role of NVCs in modulating aerosol acidity under
extreme weather conditions. These findings underscore the need to account for meteorology-
composition interactions when applying multiphase buffer theory in coastal regions, and reveal
important limitations of current models under episodic perturbations.
**2 METHODS**
**2.1 Field Measurements.** Field measurements were carried out at the Xichong site (22.48°N,
114.56°E) on the Dapeng Peninsula in Shenzhen, China, from August to September 2022. Detailed





information and site characteristics of Xichong have been described elsewhere(Zhai et al., 2025).
Briefly, Xichong is located at the southeastern end of Shenzhen, a representative coastal megacity
in southern China. During the field campaign, a Monitor for AeRosols and Gases in Ambient air
(MARGA, Metrohm-Applikon, Netherlands) was utilized to measure online concentrations of
major water-soluble gases ($NH_3$, $SO_2$, $HNO_3$, $HCl$) and aerosol ions ($NH_4^+$, $Na^+$, $K^+$, $Ca^{2+}$, $Mg^{2+}$,
$SO_4^{2-}$, $NO_3^-$, $Cl^-$). Additional measurements at the site included $PM_{2.5}$ and $O_3$ mass concentrations,
as well as key meteorological parameters (temperature, relative humidity, wind speed, and wind
direction). In this study, the analysis period used was from 24 August to 11 September 2022,
corresponding to the overlapping operational time of all deployed instruments, with all online data
standardized to a temporal resolution of 1 h.
**2.2 ISORROPIA Calculation.** The pH is defined in terms of the activity of hydrogen ions ($H^+$)
in aqueous solution, expressed on a molality basis as:
$$pH = -\log_{10}[a(H^+)] = -\log_{10}[\chi(H^+) \cdot \gamma(H^+)] \tag{1}$$

where $a(H^+)$ is the activity of $H^+$, and $\chi(H^+)$ and $\gamma(H^+)$ denote the mole fraction and mole
fraction-based activity coefficients of $H^+$ in aqueous solution, respectively. In this study, aerosol
pH is calculated using ISORROPIA II (http://isorropia.epfl.ch), which provides direct predictions
of ionic activity, gas partial pressure, and the phase volumes of solids and liquids. Here, the model
considers the $Na^+$–$NH_4^+$–$K^+$–$Ca^{2+}$–$Mg^{2+}$–$Cl^-$–$NO_3^-$–$SO_4^{2-}$ system and is run in "forward mode"
under the "metastable" phase state, which avoids salt crystallization and is more representative of
ambient aerosol conditions under high humidity. The model is driven by measured concentrations
of major aerosol ions and gases, along with observed meteorological parameters. Aerosol pH was
then calculated from the ISORROPIA output as:
$$pH = -\log_{10}[H_{aq}^+] = -\log_{10}[1000 \cdot H_{air}^+/AWC] \tag{2}$$

where $H_{aq}^+$ is the equilibrium hydronium ion concentration in ambient aerosol liquid water (mol
$L^{-1}$), $H_{air}^+$ is the equilibrium hydronium ion concentration per unit volume of air (µg m$^{-3}$), and





AWC is the aerosol water content ($\mu g\ m^{-3}$). The factor 1000 accounts for the unit conversion
between $\mu g\ m^{-3}$ of air and $mol\ L^{-1}$ of aerosol liquid water.
**2.3 Multiphase Buffer Theory.** A buffer system is defined as a chemical system that exhibits
resilience to pH perturbations upon the addition of a certain amount of acid or base. In
homogeneous aqueous solutions, the conjugate acid-base pairs are confined to the liquid phase,
and thus the solution pH is governed exclusively by acid dissociation equilibria. In multiphase
systems, however, the volatile components of the conjugate pairs can reside in both the gas and
liquid phases, whereby the pH is modulated by the coupled effects of gas-liquid partitioning and
dissociation equilibria. The recently proposed multiphase buffer theory introduced an analytical
framework for the buffer capacity $\beta$ of aerosol systems(Zheng et al., 2020). Here, $\beta$ is defined as
the amount of acid ($dn_{acid}$) required to reduce pH by $d$pH units, or the amount of base ($dn_{base}$)
required to raise pH by $d$pH units. It thus characterizes the instantaneous buffering capacity,
relating infinitesimal changes in acid/base content to the corresponding change in pH. In this study,
$\beta$ (mol $kg^{-1}$) is expressed as:
$$\beta = -\frac{dn_{acid}}{d\mathrm{pH}} = \frac{dn_{base}}{d\mathrm{pH}} = 2.303\left\{\frac{K_w}{[\mathrm{H^+}]} + [\mathrm{H^+}] + \sum_i \frac{K_{a,i}{}^* \cdot [\mathrm{H^+}]}{(K_{a,i}{}^* + [\mathrm{H^+}])^2} \cdot [X_i]_{tot}{}^*\right\} \qquad (3-1)$$
where $K_w$ is the water dissociation constant ($mol^2\ kg^{-2}$), $K_{a,i}{}^*$ represents the effective acid
dissociation constant of buffering agent $X_i$ in gas-liquid multiphase systems (mol $kg^{-1}$), and
$[X_i]_{tot}{}^*$ is the total equivalent molality of $X_i$, accounting for both gaseous and aqueous phases
(mol $kg^{-1}$).
The self-buffering capacity of water is an inherent property conferring resistance to pH changes.
Here, it is defined as:
$$\beta_{water} = 2.303\left\{\frac{K_w}{[\mathrm{H^+}]} + [\mathrm{H^+}]\right\} \qquad (3-2)$$
which becomes appreciable only under extreme acidic (pH < 2) or alkaline (pH > 12) conditions,
but is negligible within the typical aerosol pH range. In contrast, conjugate acid-base pairs (e.g.,
$NH_4^+/NH_3$, $HNO_3/NO_3^-$, $HSO_4^-/SO_4^{2-}$) dominate buffering within this range, while minor organic



acids contribute little (Zheng et al., 2020). Similar to bulk aqueous systems, the effective buffer
range ($pK_{a,i}{}^{*}$ $\pm 1$) corresponds to the pH interval over which these conjugate pairs exert substantial
buffering effects.
Previous study (Zheng et al., 2020) further introduced the dimensionless gas-liquid partitioning
constant ($K_g$) to represent the equivalent molality of gaseous species dissolved in the aqueous
phase. This formulation enables explicit treatment of gas-particle equilibria for buffering agents
such as $NH_4^+/NH_3$ and $HNO_3/NO_3^-$. Corrections for non-ideality due to elevated ionic strength can
also be incorporated through activity coefficients. Despite simplifying assumptions, such as
equilibrium thermodynamics and the neglect of organic contributions, multiphase buffer theory
provides a powerful framework for quantifying aerosol buffering mechanisms.
For the buffering agent $NH_4^+/NH_3$,

$$NH_3 + H_2O \leftrightarrow NH_4^+ + OH^- \tag{4-1}$$

$$K_g = \frac{[NH_3(g)]}{[NH_3(aq)]} = \frac{\rho_w}{H_{NH_3} \cdot R \cdot T \cdot AWC} \tag{4-2}$$

$$K_{a,NH_3}{}^{*} = \frac{[H^+(aq)] \cdot ([NH_3(aq)] + [NH_3(g)])}{[NH_4^+(aq)]} = K_{a,NH_3} \cdot \left(1 + \frac{\rho_w}{H_{NH_3} \cdot R \cdot T \cdot AWC}\right) \tag{4-3}$$

And for the buffering agent $HNO_3/NO_3^-$,

$$HNO_3 \leftrightarrow H^+ + NO_3^- \tag{5-1}$$

$$K_g = \frac{[HNO_3(g)]}{[HNO_3(aq)]} = \frac{\rho_w}{H_{HNO_3} \cdot R \cdot T \cdot AWC} \tag{5-2}$$

$$K_{a,HNO_3}{}^{*} = \frac{[H^+(aq)] \cdot ([NO_3^-(aq)])}{[HNO_3(aq) + HNO_3(g)]} = K_{a,HNO_3} \Big/ \left(1 + \frac{\rho_w}{H_{HNO_3} \cdot R \cdot T \cdot AWC}\right) \tag{5-3}$$

where $K_{a,NH_3}$ and $K_{a,HNO_3}$ are the classical dissociation constants of $NH_3$ and $HNO_3$ (mol kg$^{-1}$),
$\rho_w$ is the density of liquid water ($1 \times 10^{12}$ µg m$^{-3}$), $R$ is the universal gas constant ($8.205 \times 10^{-2}$ atm
L mol$^{-1}$ K$^{-1}$), $T$ is the absolute temperature (K), AWC is the aerosol water content (µg m$^{-3}$), and
$H_{NH_3}$ and $H_{HNO_3}$ are the Henry's law coefficients (mol kg$^{-1}$ atm$^{-1}$).





For ambient aerosols, non-ideality due to elevated ionic strength should be considered, as it
directly influences the calculation of the effective acid dissociation constant. Accounting for this
through the inclusion of activity coefficients, the expression for the effective acid dissociation
constant in a non-ideal multiphase system is given as:
$$K_{a,NH_3}^{*,ni} = K_{a,NH_3} \cdot \left(1 + \frac{\rho_w}{H_{NH_3} \cdot R \cdot T \cdot AWC}\right) \cdot \frac{\gamma_{NH_4^+}}{\gamma_{H^+}} \quad (6-1)$$
$$K_{a,HNO_3}^{*,ni} = K_{a,HNO_3} / \left(1 + \frac{\rho_w}{H_{HNO_3} \cdot R \cdot T \cdot AWC}\right) / (\gamma_{NO_3^-} \cdot \gamma_H^+) \quad (6-2)$$
Finally, the buffer capacities associated with the $NH_4^+/NH_3$ and $HNO_3/NO_3^-$ conjugate pairs are
expressed as:
$$\beta_{NH_3} = 2.303 \frac{K_{a,NH_3}^{*,ni} \cdot [H^+]}{(K_{a,NH_3}^{*,ni} + [H^+])^2} \cdot [NH_3(g) + NH_3(aq) + NH_4^+(aq)]_{tot}^* \quad (7-1)$$
$$\beta_{HNO_3} = 2.303 \frac{K_{a,HNO_3}^{*,ni} \cdot [H^+]}{(K_{a,HNO_3}^{*,ni} + [H^+])^2} \cdot [NH_3(g) + NH_3(aq) + NH_4^+(aq)]_{tot}^* \quad (7-2)$$
Multiphase buffer theory thus establishes a quantitative framework linking aerosol acidity to
multiphase equilibria. Further details of the theoretical derivation are presented in the Supporting
Information (SI, Text S1 and Table S1).
**3 RESULTS AND DISCUSSION**
**3.1 Buffer Capacity for Aerosol Multiphase Systems.** In this study, MARGA measurements
were conducted from 24 August to 10 September 2022 at the Xichong site. Meteorological
conditions, chemical compositions obtained from MARGA, and the aerosol water content and pH
values simulated by ISORROPIA throughout the sampling period are presented in Figure 1. During
the sampling period, Typhoon Ma-on (No. 2209) passed over the sampling site at 15:00 local time
on August 24, 2022. Upon its arrival, wind speed at Xichong reached 16.5 m s$^{-1}$, the highest of the
observation period. Concurrent decreases in PM$_{2.5}$, ozone, and total water-soluble ion mass
concentrations were observed, accompanied by pronounced increases in the fractional
contributions of chloride, sodium, and other NVCs (e.g., Ca$^{2+}$, Mg$^{2+}$, K$^+$) (Figure 1a–d). This





interval is hereafter referred to as the typhoon episode (blue shading in Figure 1), with all remaining periods considered non-typhoon episodes. The ISORROPIA-simulated pH values averaged 2.81 ± 0.54 during the typhoon episode and 1.59 ± 0.45 during non-typhoon periods (Figure 1e).

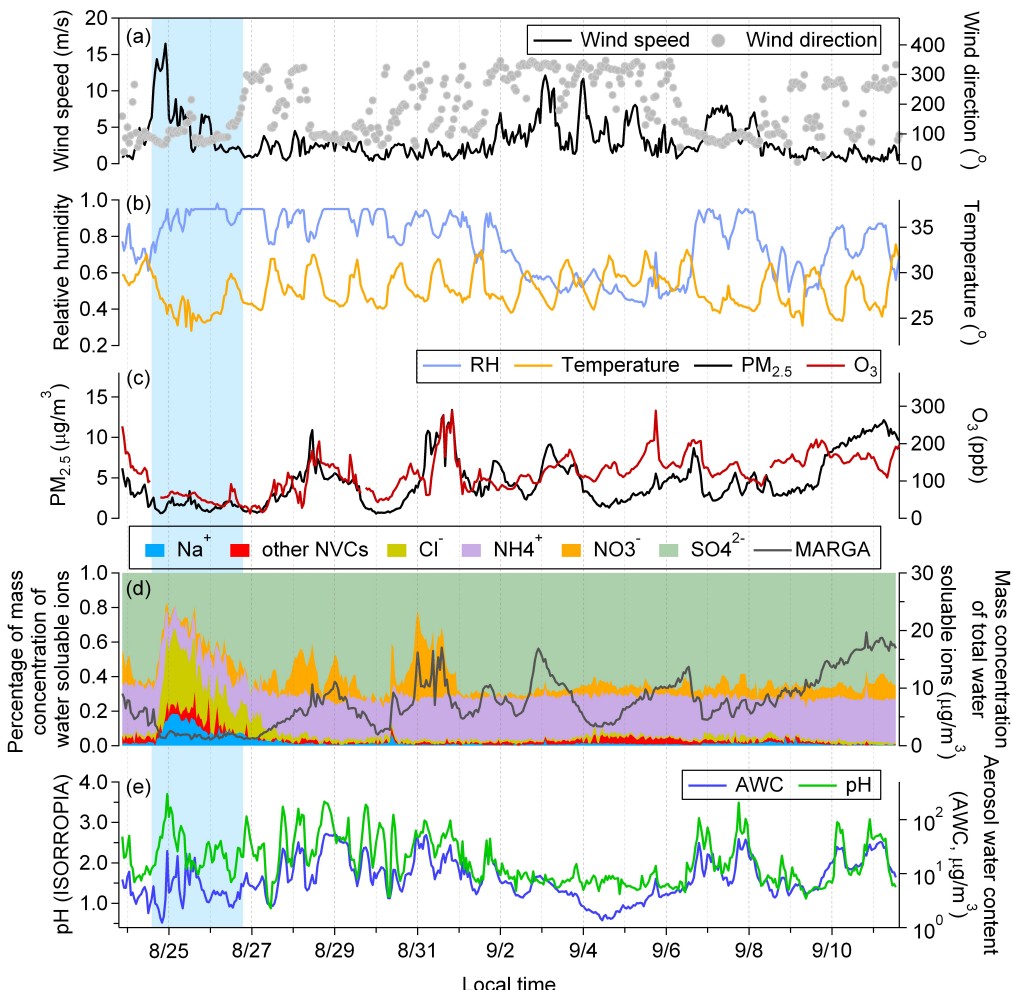

**Figure 1.** Time series of (a) wind direction and wind speed, (b) relative humidity (RH) and temperature, (c) $PM_{2.5}$ and $O_3$ concentrations, (d) aerosol composition measured by MARGA, and (e) aerosol pH and aerosol water content (AWC) simulated by ISORROPIA at the sampling site.





The gray line in panel (c) denotes the total concentration of water-soluble ions measured by
MARGA. Other NVCs include $Ca^{2+}$, $Mg^{2+}$, and $K^+$. The blue shading indicates the typhoon period.
We applied the multiphase buffer theory to calculate the buffering capacities ($\beta$) of individual
buffering agents ($NH_4^+/NH_3$, $HNO_3/NO_3^-$, and $HSO_4^-/SO_4^{2-}$) under both typhoon and non-typhoon
scenarios. In both scenarios, the largest buffering capacity was associated with the $NH_4^+/NH_3$ pair,
followed by $HSO_4^-/SO_4^{2-}$ and $HNO_3/NO_3^-$ (Figure 2). The peak buffer pH (defined as the pH
corresponding to the highest local maximum of $\beta$) for the non-typhoon scenario was ~1.62 (Figure
2a), closely matching the ISORROPIA-modeled result (1.59 ± 0.45). In contrast, the peak buffer
pH under typhoon scenario was ~1.70 (Figure 2b), showing a larger discrepancy from the
ISORROPIA-simulated result (2.81 ± 0.54).

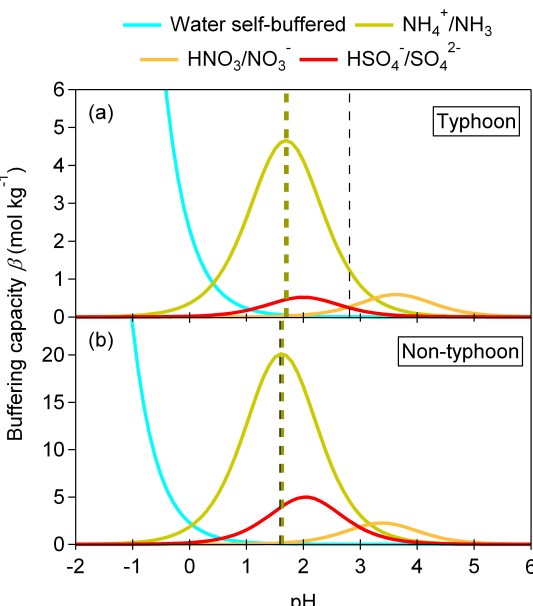


**Figure 2.** Buffering capacity ($\beta$) of the aerosol multiphase system for typhoon (a) and non-typhoon
(b) scenarios. The olive dashed line indicates the pH corresponding to the highest local maximum
of $\beta$, while the black dashed line represents the ISORROPIA-simulated pH.





The stronger role of $NH_4^+/NH_3$ under non-typhoon conditions reflects the abundance of
ammonia in this coastal environment and its efficient partitioning, which provides stable buffering
at acidic pH. However, for the typhoon scenario, the multiphase buffer theory appears to be less
applicable. It should also be noted that multiphase buffer theory assumes instantaneous
thermodynamic equilibrium and primarily considers inorganic conjugate acid-base pairs, while the
potential contributions of organic species and kinetic effects are neglected. Such simplifications
may further contribute to discrepancies under dynamic conditions.
**3.2 Contribution of Individual Drivers.** We further quantified the changes in aerosol pH ($\Delta$pH)
between typhoon and non-typhoon scenarios attributable to individual drivers (Figure 3), including
anion-normalized nonvolatile cations (NVCs) and total ammonia ($TNH_3$), the fraction of total
nitric acid ($THNO_3$) in anions, relative humidity (RH), and temperature (T). It should be noted that
the perturbation constraints imposed on each driver differ both in magnitude and physical meaning
(i.e., x-axis criteria in Figure 3, see Text S2 in SI). As a result, the $\Delta$pH contributions of different
drivers within the same scenario are not strictly comparable. Figure 3c presents the $\Delta$pH
contributions using absolute bar charts, which highlight their absolute magnitudes rather than
relative shares. In contrast, the $\Delta$pH contributions of a given driver between the typhoon and non-
typhoon scenarios are derived under the same constraint framework and can therefore be
meaningfully compared. In Figure 3, the blue and black dashed lines represent the corresponding
values for typhoon and non-typhoon scenarios, respectively. For each driver, the $\Delta$pH values
correspond to the differences between the two scenarios used as constraints. The constraint ranges
were selected to reflect the observed variability during the campaign, thereby ensuring that the
perturbations remained within realistic atmospheric conditions.
The results indicate that RH, $TNH_3$, and temperature contributed substantially more to $\Delta$pH in
the non-typhoon scenario than in the typhoon scenario, with 0.75 vs. 0.26 units for RH, 0.36 vs.
0.16 units for $TNH_3$, and 0.15 vs. 0.01 units for temperature, respectively. Notably, NVCs was the
only driver exhibiting a larger contribution to $\Delta$pH in the typhoon scenario than in the non-typhoon
scenario (0.27 vs. 0.09 units). In other words, the contribution of chemical components,

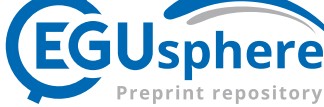



particularly NVCs, to ΔpH was more pronounced in the typhoon scenario, which partly explains
the discrepancy between the pH predicted by the multiphase buffer theory and that calculated by
ISORROPIA. This shift highlights a transition from meteorologically driven controls (RH and
temperature) under non-typhoon conditions to compositionally driven controls dominated by sea-
salt derived cations during the typhoon.

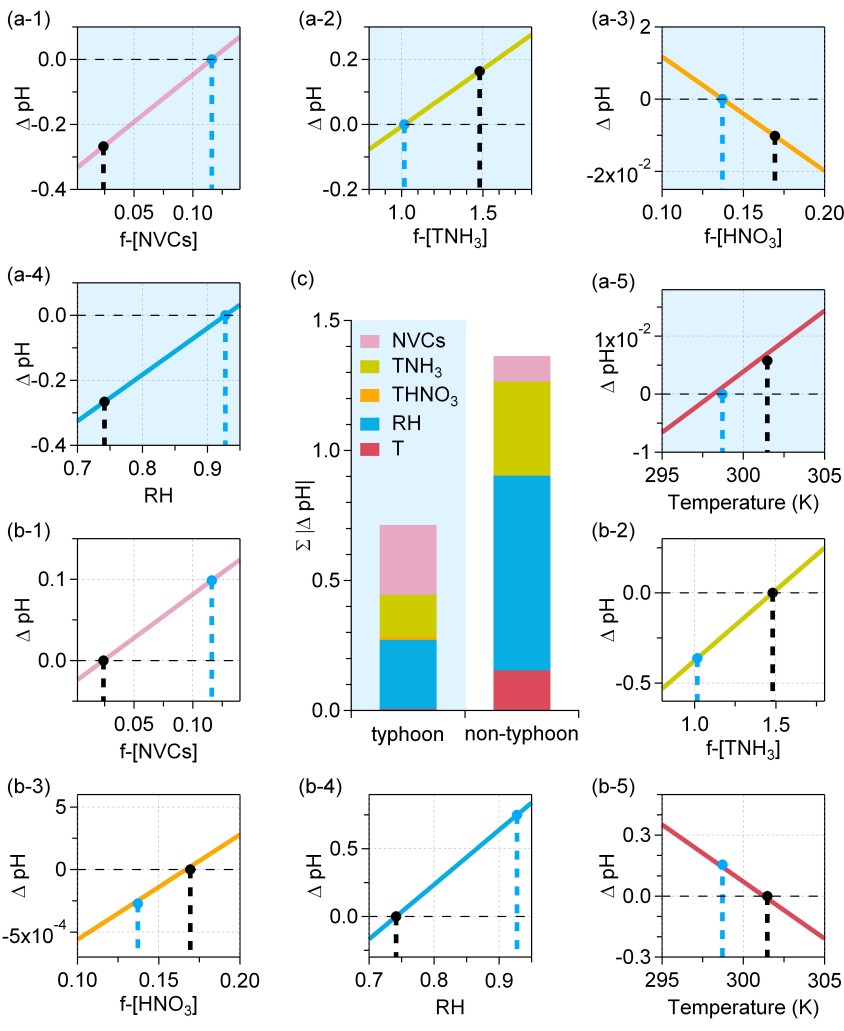




**Figure 3.** Aerosol pH differences (ΔpH) between typhoon (a1−5) and non-typhoon (b1−5) scenarios, attributed to anion-normalized non-volatile cations (NVCs) and total [NH$_3$] (TNH$_3$), the fraction of total [HNO$_3$] (THNO$_3$) in anions, relative humidity (RH), and temperature (T). The light blue background indicates simulations using typhoon conditions as inputs (a1−5), whereas the white background denotes simulations with non-typhoon conditions (b1−5). Panel (c) summarizes the total absolute ΔpH (Σ|ΔpH|) of different drivers in both scenarios. The blue and black dashed lines represent the corresponding values for typhoon and non-typhoon scenarios, respectively

Previous studies in inland regions (Zheng et al., 2020), such as the North China Plain and the southeastern United States, have emphasized ammonia availability and aerosol water as the primary determinants of aerosol acidity, whereas our results underscore the unique role of NVCs in coastal megacities under extreme weather influence. For NH$_4^+$/NH$_3$ to serve as the dominant buffering pair, two conditions must be satisfied (Zheng et al., 2020): (1) the equivalent charge of total cations exceeds that of total anions, and (2) the equivalent charge of NVCs is smaller than that of nonvolatile anions. In the typhoon scenario, however, the equivalent charge of NVCs exceeded that of nonvolatile anions, thereby violating the conditions required for NH$_4^+$/NH$_3$ to serve as the dominant buffering pair. This explains why the multiphase buffer theory fails to reliably predict aerosol pH under typhoon conditions. The enhanced influence of NVCs can be attributed to the substantial influx of sea-salt particles transported by strong winds and the altered trajectories of air masses during the typhoon. These inputs directly neutralize acidic species and disrupt the conventional NH$_4^+$/NH$_3$ buffering system, a process far less pronounced in inland settings.

We compared the ISORROPIA-simulated pH with that predicted by the multiphase buffer theory and with the ISORROPIA-simulated pH under a constant-AWC assumption, for typhoon and non-typhoon scenarios, respectively (Figure 4). The constant-AWC experiment was designed to isolate the role of chemical composition from that of aerosol water content, thereby allowing us to directly assess whether pH variability can be reproduced without explicitly accounting for





dynamic changes in AWC. The results show that under the non-typhoon scenario, ISORROPIA-
simulated pH gives a stronger correlation with the multiphase buffer theory ($R^2 = 0.69$) than with
the constant-AWC simulation ($R^2 = 0.30$). In contrast, under the typhoon scenario, ISORROPIA-
simulated pH correlates well with constant-AWC ($R^2 = 0.73$), but poorly with the multiphase buffer
theory ($R^2 = 0.21$). These findings indicate that in regimes or scenarios buffered by the $NH_4^+/NH_3$
pair, such as non-typhoon conditions in Shenzhen, variations in AWC alone can provide a reliable
prediction of aerosol pH, even without explicitly accounting for the temporal and spatial variability
in particle chemical composition. The buffering effect of ammonia suppresses the influence of
compositional differences, making aerosol water content the primary determinant of aerosol pH.
However, in environments that are not buffered by $NH_4^+/NH_3$, the influence of AWC on pH
becomes weaker, and reliable predictions require explicit consideration of chemical composition
within a constant-AWC framework.

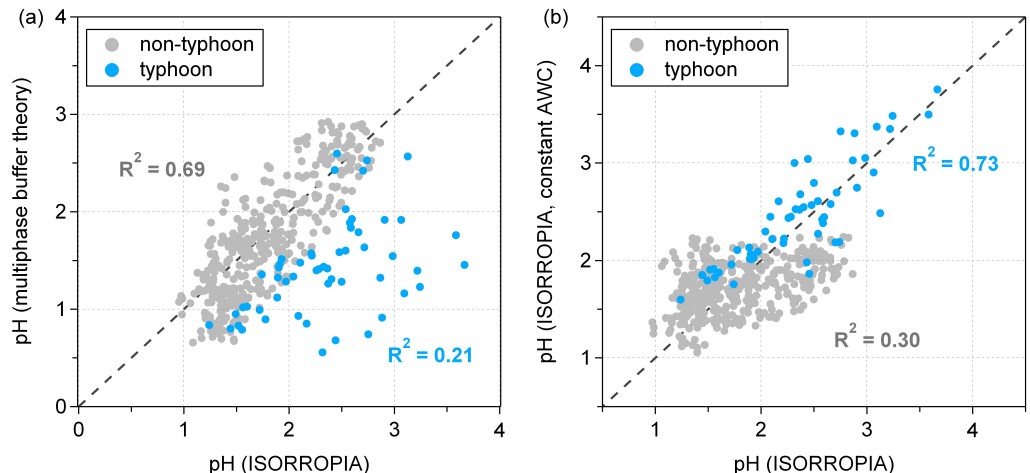


**Figure 4.** Correlations between aerosol pH simulated based on ISORROPIA and pH predicted by
(a) the multiphase buffer theory, and (b) ISORROPIA under a constant aerosol water content
(AWC) assumption with varying chemical compositions. Blue and gray dots represent typhoon
and non-typhoon scenarios, respectively. The black dashed line indicates the 1:1 reference.



These results suggest that chemical composition plays a more critical role in determining pH under typhoon scenarios, whereas during non-typhoon periods, aerosol water content influenced by RH and temperature exerts a stronger impact on pH. While the multiphase buffer theory is robust under ammonia-buffered regimes, its applicability becomes limited in environments dominated by NVCs. Refining the framework to explicitly incorporate composition-meteorology interactions is therefore essential for accurately predicting aerosol acidity in coastal regions subject to extreme weather events.

**3.3 Atmospheric implications.** In densely populated continental regions, where anthropogenic emissions and atmospheric ammonia concentrations are high, aerosol pH is likely controlled by the $NH_4^+/NH_3$ buffering pair and can therefore be reasonably approximated on the basis of aerosol mass concentration and liquid water content. This provides an opportunity to reconstruct long-term trends and large-scale spatial distributions of aerosol pH, and implies that emission reductions targeting ammonia and sulfate can exert direct and predictable influences on aerosol acidity.

In coastal regions, however, whether ammonia serves as the dominant buffering pair is strongly influenced by seasonality and meteorological conditions. Our field observations directly captured two contrasting situations: a non-typhoon period, in which ammonia acted as the dominant buffering pair, and a typhoon period, during which NVCs played a more important role. In the former case, aerosol pH can be reliably predicted using only aerosol mass concentration and AWC, whereas in the latter case, more detailed chemical composition information is required for accurate prediction. Acidity regulation in coastal area is inherently more variable than in continental settings, with rapid transitions between meteorological and compositional control. Such variability complicates the prediction of secondary aerosol formation and the assessment of pH-dependent processes (e.g., metal solubility and heterogeneous chemistry). Our results therefore underscore the need for improved representation of composition-meteorology interactions in chemical transport models and highlight the need for targeted observations during extreme weather events to constrain acidity regulation in coastal atmospheres.



**Data availability.** Data used to produce the plots within this work are available in Zenodo (https://https://zenodo.org/records/17207845).

**Author contributions.** JZ and XY designed the study. JZ and YZ analyzed the data. JZ wrote the manuscript. All co-authors contributed to discussions and suggestions in finalizing the manuscript.

**Competing interests.** The contact author has declared that none of the authors has any competing interests.

**Acknowledgments.** The authors would like to thank the Shenzhen National Climate Observatory for providing the observation platform for this study.

**Financial support.** This work was supported by the National Natural Science Foundation of China (42305108, 42530609), the Guangdong Basic and Applied Research Foundation (2025A1515011148), the Guangdong Provincial Observation and Research Station for Coastal Atmosphere and Climate of the Greater Bay Area (2021B1212050024), the Shenzhen Science and Technology Program (KQTD20210811090048025, KCXFZ20230731093601003), and the Ministry of Science and Technology of the People's Republic of China (2023YFE0112901).



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
