# Peer review of "Manuscript for # Divergent Drivers of Aerosol Acidity: Evidence for Shifting Regulatory Regimes in a Coastal Region"

_EGUsphere, 2025_

## Author Comment (AC1)

**Response to Reviewers**

We sincerely thank all anonymous referees for their valuable comments and suggestions. We have extensively revised the manuscript according to the reviewers' comments. Below, we provide a point-by-point response to each comment, along with a description of the corresponding changes made in the revised manuscript.

**Anonymous Referee #1**

The article titled "Divergent Drivers of Aerosol Acidity: Evidence for Shifting Regulatory Regimes in a Coastal Region" combines field observations, multiphase buffer theory, and ISORROPIA simulations to investigate aerosol pH dynamics in Shenzhen under typhoon and non-typhoon conditions. The work holds scientific value and practical relevance. Some issues need to be addressed.

**Response:** We sincerely thank the reviewer for the thoughtful assessment of our work. We have carefully considered all comments and revised the manuscript accordingly. Detailed point-by-point responses are provided below.

1. The Introduction could be more concise and refined. I recommend condensing the literature review and stating the novelty and contributions of this study more explicitly.

   **Response:** We thank the reviewer for this constructive suggestion. Specifically, the novelty and contributions of this study include, 1) we provide a field-based comparison of aerosol pH buffering behavior between typhoon and non-typhoon regimes in a coastal megacity, offering a natural experiment to assess acidity responses under rapidly changing atmospheric conditions; 2) we present direct observational evidence for the substantial role of NVCs in regulating aerosol acidity during extreme meteorological events, a mechanism previously suggested but rarely quantified; 3) we evaluate the applicability and limitations of multiphase buffer theory and thermodynamic models in coastal environments, where episodic perturbations challenge conventional equilibrium assumptions.

   In the revised manuscript, we have strengthened the description of the study's novelty by adding clearer statements of the scientific gaps and our main contributions as follows:

   *In this study, we integrated field observations with multiphase buffer theory and ISORROPIA II simulations to investigate the buffering capacity and controlling factors of aerosol pH in Shenzhen, China, a subtropical coastal megacity frequently influenced by typhoons. Our field observations captured both a typhoon episode and typical non-typhoon periods, providing a natural contrast between distinct atmospheric regimes. The comparison between typhoon and non-typhoon regimes enables explicit characterization of the coupled effects of aerosol composition and meteorological variability on acidity regulation under rapidly changing conditions. Moreover, our results provide direct field-based evidence for the substantial role of NVCs in modulating aerosol acidity under extreme weather conditions. These findings underscore the need to account for meteorology-composition interactions when applying multiphase buffer theory in coastal regions and reveal important limitations of current models under episodic perturbations.*

   While the first two paragraphs serve as informative background, it is also advisable to supplement with additional research reviews focusing on aerosol acidity in coastal cities.

   **Response:** We thank the reviewer's suggestion. We have added a review paragraph on aerosol acidity in coastal areas as follows:

   *Coastal and marine-influenced atmospheres represent chemically distinct environments from the continental counterparts. Nonvolatile cations derived from sea salts can modify*

*aerosol ionic balance and buffering capacity, while high relative humidity and abundant aerosol liquid water substantially affect gas-particle partitioning (Wang et al., 2022a; Wang et al., 2022b). Predicting the bulk aerosol pH and quantifying the contribution of sea salt in coastal regions is challenging due to the mixing of land and marine emissions, multiphase reactions on sea-salt surfaces, and the complexity of aerosol mixing states (Pye et al., 2020; Bougiatioti et al., 2016; Weber et al., 2016). In addition, sea-land breezes, monsoonal flows, and typhoons induce rapid shifts in aerosol sources, water content, and thermodynamic conditions, resulting in more complex acidity regulating mechanisms (Liu et al., 2019; Farren et al., 2019). Although previous studies have suggested that sea salts can substantially influence aerosol acidity in some coastal regions, direct field evidence particularly under extreme weather conditions remains scarce.*

2. Is the composition of atmospheric aerosols, as well as temperature and humidity, still affected by the typhoon for a period immediately after the typhoon ends? How can such residual effects be distinguished from or eliminated when analyzing non-typhoon events?

**Response:** We thank the reviewer for pointing this out. We agree that meteorological residuals may persist for some time after a typhoon and should be carefully excluded from non-typhoon periods. In defining the typhoon period for Typhoon Ma-on, we used both the timing of the maximum wind speed associated with the storm and the official declaration of typhoon passage issued at 19:00 on 25 August 2022. To avoid any residual influence, we further included data for the subsequent 24 hours in the typhoon-affected category.

Meteorological variables during this time confirmed the dissipation of typhoon influence: surface wind speeds remained below 3 m s$^{-1}$ for more than 20 hours, and the wind direction shifted from the typhoon-induced easterly marine flow to the regular inland background flow. These observations indicate that atmospheric conditions had stabilized before the start of the non-typhoon dataset. We have added these criteria in the revised manuscript as follows:

*The typhoon-affected period was defined based on the timing of maximum wind speed and the official declaration of typhoon passage at 19:00 on 25 August 2022. To avoid residual influence, a subsequent 24-hour period was also included in the typhoon-affected period. Meteorological observations (wind speed < 3 m s$^{-1}$ and a transition from easterly marine flow to inland background flow) confirmed that atmospheric conditions had stabilized before the start of the non-typhoon dataset.*

3. Lines 224-228, I didn't find the detailed calculation instructions on how to determine the peak buffer pH value shown in Figure 2. Is the curve shown in Figure 2 calculated based on the average of all typhoon and non-typhoon scenarios? More clarity is needed for the relevant description.

**Response:** Thank you for pointing out the need for clarification. The buffer curves shown in Figure 2 were calculated based on the average chemical compositions of the typhoon and non-typhoon scenarios, respectively. These averaged input concentrations were then used to derive the corresponding buffer capacities of the individual buffering agents. The peak buffer pH, defined as the pH corresponding to the highest local maximum of $\beta$, is directly obtained from the calculated buffer capacity curves.

To improve clarity, we have added the following description in the revised manuscript:

*We applied the multiphase buffer theory to calculate the buffering capacities ($\beta$) of individual buffering agents ($NH_4^+/NH_3$, $HNO_3/NO_3^-$, and $HSO_4^-/SO_4^{2-}$) under both typhoon and non-typhoon scenarios. The $\beta$ values were calculated using the average chemical*

*compositions of the typhoon and non-typhoon periods, respectively, with the averaged inputs summarized in Table S2. In both scenarios, the $NH_4^+/NH_3$ pair exhibited the greatest buffering capacity, followed by $HSO_4^-/SO_4^{2-}$ and $HNO_3/NO_3^-$ (Figure 2). The peak buffer pH (defined as the pH corresponding to the local maximum of $\beta$) for the non-typhoon scenario was ~1.62 (Figure 2a), closely matching the ISORROPIA-modeled result (1.59 ± 0.45).*

4. Line 240, the method to quantify the changes in aerosol pH (ΔpH) between typhoon and non-typhoon scenarios attributable to individual drivers should be presented in detail in the main text. How to obtain the result shown in Figure 3?

**Response:** We thank the reviewer for this comment. ΔpH for an individual driver was calculated as the difference between the simulated pH values obtained by constraining the ISORROPIA model with typhoon and non-typhoon datasets, while perturbing only the driver of interest within its observed variability range and holding all other inputs constant. For each driver, ΔpH was obtained by substituting the typhoon and non-typhoon driver values into the same thermodynamic setting while holding all other inputs constant.

To illustrate the calculation, we take relative humidity (RH) as an example. Each scenario has its own thermodynamic input setting in ISORROPIA: $S_t$ for the typhoon period and $S_{nt}$ for the non-typhoon period. RH also has two corresponding values: $RH_t$ (typhoon) and $RH_{nt}$ (non-typhoon). To isolate the contribution of RH, we substituted both values into the same constraint setting while keeping all other inputs constant.

Under the typhoon input setting $S_t$, we calculated:

$$\Delta pH^{\text{typhoon}}(RH) = |pH(S_t, RH_t) - pH(S_t, RH_{nt})|$$

This represents how much the pH in the typhoon environment would change if RH were replaced by its non-typhoon value.

Similarly, under the non-typhoon constraint setting $S_{nt}$, we calculated:

$$\Delta pH^{\text{non-typhoon}}(RH) = |pH(S_{nt}, RH_t) - pH(S_{nt}, RH_{nt})|$$

This approach isolates the effect of each driver by applying its typhoon and non-typhoon values within an identical constraint framework. We revised the manuscript as follows:

*We further quantified the changes in aerosol pH (ΔpH) between typhoon and non-typhoon scenarios attributable to individual drivers (Figure 3), including anion-normalized nonvolatile cations (NVCs) and total ammonia (TNH₃), the fraction of total nitric acid (THNO₃) in anions, relative humidity (RH), and temperature (T). It should be noted that the perturbation constraints imposed on each driver differ both in magnitude and physical interpretation (i.e., x-axis criteria in Figure 3, see Text S2 in SI). As a result, the ΔpH contributions of different drivers within the same scenario are not strictly comparable. Specifically, ΔpH was calculated as the difference between the simulated pH values obtained by constraining the ISORROPIA model with typhoon and non-typhoon datasets, while perturbing only the driver of interest within the observed variability range and holding all other inputs constant. In contrast, the ΔpH contributions of a given driver between the typhoon and non-typhoon scenarios are derived under the same constraint*

*framework and can therefore be meaningfully compared. In Figure 3, the blue and black dashed lines represent the corresponding values for typhoon and non-typhoon scenarios, respectively. For each driver, the ΔpH values correspond to the pH differences between the two scenarios under the same constraints. The constraint ranges were selected to reflect the observed variability during the campaign, thereby ensuring that the perturbations remained within realistic atmospheric conditions. Figure 3c presents the ΔpH contributions using absolute bar charts, which highlight their absolute magnitudes rather than relative shares. While the buffering capacity β (Figure 2) describes the intrinsic pH sensitivity within a single scenario, the ΔpH in Figure 3 quantifies the driver-induced pH difference between typhoon and non-typhoon scenarios under the same perturbation constraints. Therefore, β and ΔpH represent different dimensions of aerosol acidity.*

5. Lines 288-290, I am unclear about what the pH values being compared in Figure 4 actually mean. The description here is a bit confusing and not clear enough.

   **Response:** Thank you for this comment. Here, we conducted this set of comparisons to further quantify the relative contributions of aerosol water content (AWC) and chemical composition to aerosol pH variability. The combination of observationally constrained buffer theory pH, standard ISORROPIA simulations, and constant-AWC experiments provides a diagnostic framework that isolates the roles of AWC and composition under contrasting meteorological conditions.

   "ISORROPIA-simulated pH" refers to the standard model output using measured temperature, RH, and aerosol chemical composition. "Buffer theory pH" refers to pH diagnosed from observations using the multiphase buffer theory based on measured ion concentrations. "Constant-AWC pH" refers to the ISORROPIA-simulated pH under a fixed aerosol water content, designed to isolate the influence of chemical composition.

   We have added a more detailed description in the revised manuscript as follows:

   *We compared the ISORROPIA-simulated pH with both the pH predicted by multiphase buffer theory and the ISORROPIA-simulated pH under a constant AWC assumption for typhoon and non-typhoon scenarios, respectively (Figure 4). Here, ISORROPIA-simulated pH refers to the standard model output using measured temperature, RH, and aerosol chemical composition. Multiphase buffer theory pH refers to the pH diagnosed from observations using the multiphase buffer theory based on measured ion concentrations. Constant-AWC pH corresponds to the ISORROPIA-simulated pH under a fixed aerosol water content of 10 μg m$^{-3}$ (mean AWC of all calculated conditions). The constant-AWC experiment was designed to isolate the role of chemical composition from that of aerosol water content, thereby allowing us to directly assess whether pH variability can be reproduced without explicitly accounting for dynamic changes in AWC.*

6. The unit formats used in the manuscript are inconsistent and should be standardized.

   **Response:** We thank the reviewer for pointing this out. We carefully re-examined the manuscript and identified several minor inconsistencies in unit formatting, including the use of m s$^{-1}$ vs. m/s for wind speed, μg m$^{-3}$ vs. μg/m$^3$ for concentration units. We have revised the manuscript to ensure consistency in all unit formats.

**Anonymous Referee #2**

The authors conducted observations at the coastal Xichong site in Shenzhen, integrated field data, multiphase buffer theory, and ISORROPIA simulations to comparatively analyze the buffering capacity and driving factors of aerosol pH during typhoon and non-typhoon periods. They found that during non-typhoon periods, aerosol pH is buffered by the NH4+/NH3 pair with relative humidity as the main driver, while during typhoon periods, it is dominated by sea salt-derive NVCs.

This study points out the limitations of the multiphase buffer theory in scenarios dominated by NVCs, which is innovative. The overall quality of the study is relatively high, and after addressing the following comments, I believe it meets the quality requirements of Atmospheric Chemistry and Physics.

**Response:** We sincerely appreciate the reviewer's thoughtful comments. We have carefully revised the manuscript in response to all suggestions, which we believe have further improved the quality of the study. Our detailed point-by-point responses are provided below.

The detailed comments are as follows.

1. What does "constant aerosol water" in the Abstract mean? Why is this parameter considered?

   **Response:** "Constant aerosol water" refers to a controlled sensitivity experiment in which the aerosol liquid water (ALW) predicted by ISORROPIA is artificially fixed at a representative value rather than allowed to respond dynamically to changes in relative humidity and chemical composition. We include this experiment to disentangle the effects of ALW from those of aerosol chemical composition.

   Because ALW strongly influences aerosol pH through dilution, ion activity, and gas-particle partitioning, variations in ALW often dominate pH changes in humid subtropical environments. By holding ALW constant, we can evaluate how much of the observed pH variability can be attributed to chemical composition alone, independent of concurrent changes in water content. This diagnostic approach allows us to quantify the relative roles of ALW and composition under contrasting meteorological regimes.

   We have revised the sentence in the Abstract as follows:

   *ISORROPIA simulations under the assumption of constant aerosol water content reproduced the observed pH more reliably, highlighting a compositional rather than meteorological control.*

2. The Introduction lacks a review of how the multiphase buffer theory explains the driving factors of pH.

   **Response:** We thank the reviewer for pointing this out. We have added a review in the Introduction to clarify how multiphase buffer theory explains the driving factors of aerosol pH as follows:

   *The recently proposed multiphase buffer theory provides a new framework for understanding the evolution of aerosol acidity (Zheng et al., 2020). It demonstrates that conjugate acid-base pairs act as efficient buffers that stabilize particle pH against external*

*perturbations such as changes in emissions or sulfate production (Zheng et al., 2022b). Building on the buffering framework, it enables robust aerosol pH retrievals in ammonia-buffered regions lacking comprehensive chemical measurements (Zheng et al., 2022a). In addition, it provides a mechanistic basis for understanding the thermodynamics of displacement reactions in which strong acids or bases are substituted by weaker ones within aerosols (Chen et al., 2022). Recent study further shows that multiphase buffering constrains aerosol pH and thereby regulates the dominant aqueous sulfate formation pathways (Gao et al., 2025).*

3. This study uses ISORROPIA to calculate the acidity of inorganic aerosols, and it is necessary to define the boundaries at an appropriate position.

**Response:** Thank you for this helpful comment. In the revised manuscript, we have added descriptions acknowledging that ISORROPIA resolves only the inorganic thermodynamic system and does not explicitly treat organic aerosol components as follows:

*It should be noted that ISORROPIA resolves the inorganic thermodynamic system and does not explicitly treat organic aerosol components in this study. Although the direct contribution of organics to aerosol pH is relatively minor (Guo et al., 2015), interactions between inorganic and organic components can alter acidity with pH increases of up to ~0.7 units (Pye et al., 2018). Therefore, neglecting organic-inorganic interactions introduces uncertainties, but the inorganic-only framework is expected to capture the dominant acid-base chemistry governing aerosol acidity in this coastal environment.*

4. Figure 1: It is suggested to use arrows to indicate wind direction instead of degrees.

**Response:** We thank the reviewer for this suggestion. We agree that wind vectors can provide an intuitive visualization of wind direction. However, our dataset has a relatively high temporal resolution, which results in a large number of data points over short time periods. When plotted as arrows, the vectors become visually dense and overlap substantially as shown below, making the directionality difficult to interpret.

The original degree-based wind direction representation provides much clearer trends. Therefore, we retain the original format in Figure 1.

[Figure]

[Figure]

5. Figure 2: Based on the calculations using the multiphase buffer theory, can it be indicated that the buffer peak is the range where the pH value should lie? Is there evidence from other studies to support this, or is it a coincidental result of this paper?

**Response:** Thank you for this important question. According to the multiphase buffer theory, the buffer peak represents the pH at which the buffering capacity of a given acid-base pair is maximized. This indicates a chemically stable pH regime where the system is least sensitive to perturbations, but it does not imply that the ambient aerosol pH must lie exactly at the peak.

Previous studies (Zheng et al., 2020; Zheng et al., 2022a; Zheng et al., 2022b; Zhou et al., 2022; Gao et al., 2025; Zheng et al., 2024; Zheng et al., 2023) have shown that ambient aerosol pH often lies near the dominant buffer region, because the prevalent inorganic composition constrains the acidity within a relatively stable range.

In our non-typhoon scenario, the observed aerosol pH indeed lies close to the dominant $NH_4^+/NH_3$ buffer regime, consistent with the multiphase buffer theory and previous studies. However, under typhoon conditions, this theoretical framework becomes less applicable. We attribute this deviation primarily to the substantial input of non-volatile

cations (NVCs, e.g., $Na^+$, $Ca^{2+}$, $Mg^{2+}$) transported from marine and coastal sources by the typhoon circulation, as demonstrated by our quantitative analysis of the contribution of individual drivers (Section 3.2). These NVCs may neutralize acidic species, modify ion activity ratios, and effectively shift the acid-base balance away from the typical ammonium-buffered regime, leading to pH values that do not align with the expected buffer peak.

6. Do Figure 2 and Figure 3 contradict each other? The former indicates that the buffer capacity during non-typhoon periods is stronger, so why is the ΔpH here much larger than that during typhoon periods? This is especially considering that the difference in the standard deviation of the data reported in Figure 1 is not significant.

**Response:** Thank you for this comment. Figure 2 shows the buffering capacity ($\beta$), which describes the intrinsic ability of a given acid-base pair (e.g., $NH_4^+/NH_3$, $HSO_4^-/SO_4^{2-}$) to resist pH perturbations at a given chemical composition. In other words, $\beta$ represents how sensitive the pH is to the addition or removal of acidity, and it is a property of the thermodynamic system at that state. The higher $\beta$ during non-typhoon periods simply indicates that the aerosol system is more strongly buffered (i.e., pH is less responsive to perturbations).

In contrast, Figure 3 shows the ΔpH, which quantifies how much each individual driver (e.g., NVCs, $TNH_3$, $THNO_3$, RH, T) contributes to the pH difference between typhoon and non-typhoon scenarios under the same perturbation constraints. In other words, ΔpH represents the net contribution difference of each driver between the two scenarios, not the intrinsic buffering strength within a single scenario.

Because $\beta$ and ΔpH describe different dimensions of the system, one is sensitivity within a scenario (Figure 2), and the other is driver-induced pH change between scenarios (Figure 3), a larger ΔpH in Figure 3 does not contradict the stronger buffering capacity shown in Figure 2 for non-typhoon periods. The two figures address complementary aspects of aerosol acidity behavior. To improve clarity, we added some explanation of in the revised manuscript as follows:

*While the buffering capacity $\beta$ (Figure 2) describes the intrinsic pH sensitivity within a single scenario, the ΔpH in Figure 3 quantifies the driver-induced pH difference between typhoon and non-typhoon scenarios under the same perturbation constraints. Therefore, $\beta$ and ΔpH represent different dimensions of aerosol acidity.*

7. Figure 4: What previous evidence led the authors to want to fix AWC? And to what value is AWC fixed? The mean? The median?

**Response:** Thank you for raising this important point. Our motivation for fixing AWC stems from that aerosol pH is often strongly controlled by aerosol water content, particularly in regimes influenced by meteorology or hygroscopic growth, suggesting that under certain atmospheric conditions, variations in AWC alone can reproduce most of the observed pH variability, whereas the role of chemical composition becomes comparatively suppressed. Our constant-AWC experiment follows the methodological approach of Zheng et al. (2020), who showed that fixing AWC allows the isolated evaluation of compositional controls on aerosol pH.

In our constant-AWC experiment, AWC was fixed to the campaign-mean value for 10 μg m$^{-3}$. We have now clarified the motivation and the fixed AWC values in the revised manuscript.

*We compared the ISORROPIA-simulated pH with both the pH predicted by multiphase buffer theory and the ISORROPIA-simulated pH under a constant AWC assumption for typhoon and non-typhoon scenarios, respectively (Figure 4). Here, ISORROPIA-simulated pH refers to the standard model output using measured temperature, RH, and aerosol chemical composition. Multiphase buffer theory pH refers to the pH diagnosed from observations using the multiphase buffer theory based on measured ion concentrations. Constant-AWC pH corresponds to the ISORROPIA-simulated pH under a fixed aerosol water content of 10 μg m$^{-3}$ (mean AWC of all calculated conditions). The constant-AWC experiment was designed to isolate the role of chemical composition from that of aerosol water content, thereby allowing us to directly assess whether pH variability can be reproduced without explicitly accounting for dynamic changes in AWC.*

**References:**

Gao, J., Wei, Y. T., Wang, H. Q., Song, S. J., Xu, H., Feng, Y. C., Shi, G. L., and Russell, A. G.: Multiphase Buffering: A Mechanistic Regulator of Aerosol Sulfate Formation and Its Dominant Pathways, Environ. Sci. Technol., 59, 8073–8084, 10.1021/acs.est.4c13744, 2025.

Zheng, G., Su, H., and Cheng, Y.: Role of Carbon Dioxide, Ammonia, and Organic Acids in Buffering Atmospheric Acidity: The Distinct Contribution in Clouds and Aerosols, Environ. Sci. Technol., 10.1021/acs.est.2c09851, 2023.

Zheng, G., Su, H., Andreae, M., Pöschl, U., and Cheng, Y.: Multiphase Buffering by Ammonia Sustains Sulfate Production in Atmospheric Aerosols, Agu Advances, 5, 10.1029/2024av001238, 2024.

Zheng, G., Su, H., Wang, S., Pozzer, A., and Cheng, Y.: Impact of non-ideality on reconstructing spatial and temporal variations in aerosol acidity with multiphase buffer theory, Atmospheric Chemistry and Physics, 22, 47–63, 10.5194/acp-22-47-2022, 2022a.

Zheng, G. J., Su, H., and Cheng, Y. F.: Revisiting the Key Driving Processes of the Decadal Trend of Aerosol Acidity in the U.S, Acs Environmental Au, 2, 346–353, 10.1021/acsenvironau.1c00055, 2022b.

Zheng, G. J., Su, H., Wang, S. W., Andreae, M. O., Pöschl, U., and Cheng, Y. F.: Multiphase buffer theory explains contrasts in atmospheric aerosol acidity, Science, 369, 1374–+, 10.1126/science.aba3719, 2020.

Zhou, M., Zheng, G. J., Wang, H. L., Qiao, L. P., Zhu, S. H., Huang, D. D., An, J. Y., Lou, S. R., Tao, S. K., Wang, Q., Yan, R. S., Ma, Y. G., Chen, C. H., Cheng, Y. F., Su, H., and Huang, C.: Long-term trends and drivers of aerosol pH in eastern China, Atmospheric Chemistry And Physics, 22, 13833–13844, 10.5194/acp-22-13833-2022, 2022.